# Hematological Profile of Pregnant Women with Suspected Zika Virus Infection Followed Up at a Referral Service in Manaus, Brazil

**DOI:** 10.3390/v13040710

**Published:** 2021-04-20

**Authors:** Anny Beatriz Costa Antony de Andrade, Maria Jacirema Ferreira Gonçalves, Elijane de Fátima Redivo, Maria das Graças Costa Alecrim, Flor Ernestina Martinez-Espinosa

**Affiliations:** 1Oswaldo Cruz Foundation, Leônidas & Maria Deane Institute, Amazonas 69057-070, Brazil; antony.beatriz@gmail.com; 2Nursing College, Federal University of Amazonas, Amazonas 69057-070, Brazil; jacirema.goncalves@gmail.com; 3Tropical Medicine Foundation Dr. Heitor Vieira Dourado, Amazonas 69040-000, Brazil; elijaneredivo@hotmail.com (E.d.F.R.); galecrim.br@gmail.com (M.d.G.C.A.)

**Keywords:** arboviruses, Zika virus, pregnancy, maternal health

## Abstract

The purpose of this paper is to describe the hematological profile of pregnant women with suspected Zika virus (ZIKV) infection followed up at a reference service for infectious diseases in Manaus, Brazil, through a clinical, epidemiological, cross-sectional study of pregnant women with an exanthematic manifestation who looked for care between 2015 and 2017. The participants were 499 pregnant women, classified into four subgroups, according to laboratory confirmation of infections: ZIKV-positive; ZIKV-positive and positive for another infection; positive for another infection but not ZIKV-positive; and not positive for any of the infections investigated. Hematological parameters were analyzed descriptively. The association between maternal infection and the hematological profile, along with the association between the maternal hematological profile and the gestational outcome, were tested. Similar hematic and platelet parameters were observed among pregnant women. However, a significant association was observed between low maternal lymphocyte count and a positive diagnosis for ZIKV (*p* < 0.001). The increase in maternal platelet count and the occurrence of unfavorable gestational outcome were positively associated. A similar hematic and platelet profile was identified among pregnant women, differing only in the low lymphocyte count among ZIKV-positive pregnant women. Regarding gestational outcomes, in addition to the damage caused by ZIKV infection, altered maternal platelets may lead to unfavorable outcomes, with the need for adequate follow-up during prenatal care.

## 1. Introduction

Zika virus (ZIKV) infection is a 21st century health emergency, which was declared as such by the World Health Organization [1]. It is associated with altered development in children exposed to infection during the maternal gestational period [2]. Since the beginning of the ZIKV epidemic in 2015 in northeastern Brazil, the consequences of congenital infection have remained under investigation [3].

The relationship between ZIKV infection during pregnancy and the occurrence of deleterious outcomes should take into account changes in the maternal condition that might affect the maintenance of the pregnancy [4,5]. Modulation of the maternal immune system occurs in order to prevent the pregnancy from being interrupted by the cytotoxicity of lymphocytes, which can cause a prolonged response to diseases, even with an increase in the number of cells in the innate immune system [4,6,7].

In addition to changes inherent to the gestational period, such as hemodilution and thrombocytopenia [8,9,10], hematological changes can be triggered by infections. Erythropenia, leukopenia, and thrombocytopenia have been observed in pregnant women affected by other arboviruses, such as the Dengue (DENV) and Chikungunya (CHIKV) viruses [11,12]. Hematological alterations as a result of ZIKV have been described in the general population, and findings include thrombocytopenia and leukocytosis [13,14,15,16,17,18]. Only one prospective study, conducted on pregnant non-human primate females, found the occurrence of maternal neutrophilia [19].

Observing the hematological alterations that may occur during pregnancy and those resulting from arbovirus infection, there is a need to describe the hematological characteristics of pregnant women with ZIKV, considering the changes in the gestational period. The findings will elucidate possible changes in the maternal hematological system triggered by ZIKV infection during pregnancy, and their repercussions with respect to gestational outcomes.

The aim of this study is to describe the hematological profile of pregnant women with suspected ZIKV infection followed up in a reference service for infectious diseases in Manaus, the capital of Amazonas, in northern Brazil.

## 2. Materials and Methods

A cross-sectional, clinical, epidemiological study was conducted, which was linked to a cohort study on pregnant women with exanthematic status conducted by the Foundation of Tropical Medicine Doctor Heitor Vieira Dourado (FMT-HVD) during the most intense period of transmission of ZIKV in Amazonas.

FMT-HVD is a worldwide reference center for the treatment of acute, infectious, parasitic, and dermatological febrile diseases. The institution treats patients from Amazonas and other Brazilian states and neighboring countries.

About 860 pregnant women sought care at FMT-HVD between 2015 and 2017, and were referred to the Infectious and Parasitic Diseases (DIP) outpatient clinic for follow-up with a multidisciplinary team.

The pregnant women included in the present study did not present a history of autoimmune diseases or hematological disorders.

Socio-demographic and obstetric data were collected during the first visit of the pregnant women. When available, the information described in the prenatal report was collected. Hematological data, as well as data related to infections, were obtained through analyses of urine and blood samples. Neonatal data were collected during puerperal consultations with women who were followed up through outpatient care.

For blood samples, 10 mL was collected through venipuncture on the forearm of each pregnant woman using a system with a sterile and disposable needle. Each sample was stored in a sterile tube (an EDTA blood tube without additives) and subsequently analyzed. For urinary samples, 50 mL of urine was collected in a sterile vial within 12 days of the appearance of exanthematic signs.

Blood samples were preliminarily tested for agents involved in TORCH syndrome, which is a cluster of symptoms and clinical signs caused by congenital infection of viruses such as the rubella virus (Rubella IgM/IgG—Chemiluminescence), cytomegalovirus (Cytomegalovirus IgM/IgG—Chemiluminescence), herpes simplex (Herpes IgM/IgG), and parvovirus (Parvovirus B19 IgM serological kit), as well as other agents such as toxoplasma (IgM/IgG—Chemiluminescence) and syphilis (Chemiluminescence Immunoassay). These were all investigated, in addition to chronic infections of hepatitis B (RT-PCR and HBsAg— Enzyme-Linked Immunosorbent Assay (ELISA)) and C (anti–HCV—ELISA). The presence of HIV infection was investigated using an immuno-enzymatic assay (anti-HIV—ELISA I).

For the diagnosis of ZIKV, blood and urine samples were processed through a reverse transcriptase reaction, followed by a real-time polymerase chain reaction (RT-PCR) [20], a method employed by the Central Laboratory of Public Health (LACEN) of Manaus and the Laboratory of Clinical Analysis (LAC) of FMT-HVD, which were the laboratories responsible for performing the ZIKV test. A diagnosis of ZIKV was confirmed by the detection of viral RNA fragments in one or both samples.

Diagnosis of infection by the other arboviruses (Dengue, Chikungunya, Mayaro, Oropouche) was confirmed through the detection of viral RNA through RT-PCR, and if the patient had been manifesting the disease for five or more days, the samples were evaluated by immuno-enzymatic assays (Dengue IgM serological kit; Chikungunya: MAC- ELISA; Mayaro and Oropouche: immunoenzymatic method using infected cells as antigens (EIA-ICC) ELISA) [21,22,23,24]. Tests were not performed if the patient had recently had tests performed during prenatal care, which had already been performed at LACEN during the prenatal routine.

After data collection, a database was built using Microsoft Excel 2016 software. Data analysis was processed using Software RStudio version 1.3.959.

For the purpose of analysis, four subgroups of pregnant women were established according to the type of infection: positive Zika virus (positive diagnostic test result only for ZIKV), positive Zika virus and other infection (positive diagnostic test result for ZIKV and for other infections), another infection (positive diagnostic test result for other infections except ZIKV), and not positive (negative or indeterminate diagnostic test result for ZIKV and negative for the other infections investigated).

For data analysis, variable gestational outcomes were created, considering as unfavorable outcomes the presence of any of the following conditions: miscarriage, fetal and/or neonatal death, prematurity, low birth weight, newborn (NB) with microcephaly, and NB with immediate neonatal adaptation by an Apgar score of ≤7 points (a test given at 1 min after birth to check a baby’s heart rate, muscle tone, and other signs to see if extra medical care or emergency care is needed). The presence of all of the following characteristics was considered a favorable outcome: birth of a living conceptus from 37 weeks on, birth weight ≥ 2500 g, no microcephaly or other malformations, and an Apgar score ≥ 8 points during the first minute of life.

The association of infection (ZIKV, ZIKV and other infection, other infections except ZIKV, and non-positive for any infection) with hematological variables was tested using a bivariate Analysis of variance (ANOVA). For variables that did not fit the ANOVA, the Kruskal–Wallis nonparametric test was used, followed by Dunn’s post-hoc test, for multiple comparisons of the medians and interquartile intervals presented for each infection situation. The statistical significance considered was *p*-value < 5%.

Considering the variable gestational outcome as dichotomous, namely an unfavorable or favorable gestational outcome, the association with hematological variables was verified through bivariate analysis with Student’s t test. For the variables that did not fit this test, the Mann–Whitney test was used. Variables with *p*-value < 0.2 were included in the crude log-binomial regression model, inserted one by one, evaluating the significance and the magnitude of the measurements. The relative risk estimate was used, leaving only the variables with statistical significance below 5% in the adjusted final model. In the modeling process, there was no interaction or effect modification detected. The quality of the model adjustment was based on the half-normal plot with simulated envelopes.

The present study was approved in the Research Ethics Committee of the FMT-HVD under protocol code: 60168216.2.0000.0005, Process Number 2.375.813.

## 3. Results

Figure 1 shows the recruitment flow of pregnant women in this study.

Among the 499 women who met the eligibility criteria of the study, 212 were diagnosed with ZIKV, of whom 166 (33.2%) were positive only for ZIKV, while the other 46 (9.2%) patients were diagnosed with ZIKV and other infections. There were 57 (11.4%) patients diagnosed with other infections excluding ZIKV, and there were 230 (46.0%) non-positive results for ZIKV or other infections investigated.

The time between the manifestation of the first symptoms and the performance of the tests, namely, the RT-PCR test and the complete blood count, ranged from 0 to 71 days, with a median of 3 days (Q1 = 2 days; Q3 = 4 days). In most cases, when the test was performed after 15 days, there was a non-positive result for ZIKV or other infections.

The majority of the pregnant women participating in the study were aged 20 to 34 years, had completed secondary school, were living with their partners, and were employed. Regarding obstetric characteristics, most of the women were primiparous in their second trimester of pregnancy. The median gestational age of manifestation of the symptoms was 22.1 weeks (Q1 = 15.1 weeks; Q3 = 30.3 weeks). There were no statistically significant differences between the socio-demographic and obstetric profiles of pregnant women according to the infection status presented. A history of abortion was reported by four (2.4%) pregnant women with ZIKV infection, and one pregnant woman with other infections. Reports of premature uterine contractions were more frequent among ZIKV-positive pregnant women (12.0%).

Regarding neonatal characteristics, weight, height, and anthropometric measurements did not present significant differences compared to infection status. The median gestational age at delivery was 39 weeks (Q1 = 38 weeks; Q3 = 40 weeks), and median birth weight, head circumference, and height were 3300 g (Q1 = 2975 g; Q3 = 3600 g), 34 cm (Q1 = 33 cm; Q3 = 35 cm), and 48 cm (Q1 = 42 cm; Q3 = 50 cm), respectively.

The occurrence of prematurity, low birth weight, microcephaly, and Apgar scores of ≤7 points during the first minute of life was higher among neonates of ZIKV-positive pregnant women, compared to other pregnant women with different infection status. Table 1 shows the data related to the gestational outcome according to infection status.

Regarding the hematological characteristics presented by the pregnant women, the median and interquartile intervals of the blood components—namely, the hematimetric and platelet indices—did not present statistically significant differences according to the women’s infection status. However, there was a significant association between the presence of ZIKV infection and altered lymphocyte values (*p* < 0.001), using non-positive pregnant women as a reference (Table 2).

There were 198 cases of hypochromia, 30.3% of which were in ZIKV-positive pregnant women. There were 145 cases of leukopenia related to neutropenia and lymphocytopenia, 37.2% of which were in ZIKV-positive pregnant women. There were 76 cases of Mean platelet volume (MPV) observed below the expected value for pregnancy, 63.1% of which were in pregnant women with ZIKV infection. The results observed did not show significant differences (data not shown in Table 2).

There were significant associations between the values of mean corpuscular volume, platelet count, procalcitonin, and unfavorable gestational outcomes (miscarriage, fetal and/or neonatal death, prematurity, low birth weight, microcephaly, and immediate neonatal adaptation by an Apgar score of ≤ 07 points during the first minute of life), as shown in Table 3.

There was a statistically significant association between platelet count and unfavorable outcomes of pregnancy. The increase in platelet count was associated with the risk for unfavorable gestational outcomes, although there was low magnitude in this measure (Table 4).

## 4. Discussion

The similarities in the maternal socio-demographic characteristics between the groups regarding the presence of ZIKV infections should be interpreted as non-differential, since the infectious agent does not act according to social class, despite the recognition of the impacts that infection can have on populations with precarious conditions and greater vulnerability to the infectious agent [25,26].

The observed profile of young adult women who live with their partners and with higher educational attainment is similar to that found in women with more than seven prenatal care visits [27]. The level of clarification presented by the pregnant women may have helped in the early identification of their symptoms and prompted their immediate search for care in the healthcare system, influencing their time of follow-up and reducing deleterious outcomes to the mother–child binomial.

The concentration of patients who manifested symptoms from the second trimester of pregnancy may be related to the occurrence of asymptomatic [28] or oligosymptomatic cases during other gestational periods in other women, which may have decreased their likelihood of actively searching for the referral service and contributed to the pregnant women remaining unaware of the infection’s severity.

A similar period of symptom manifestation was observed in longitudinal studies developed with ZIKV-positive pregnant women [29,30,31]. ZIKV infection during the second gestational trimester may increase the chances of altered development of the conceptus by 5% [32].

The history of abortion reported by pregnant women may have influenced the outcome of some pregnancies in progress during the study. A history of previous abortions leads to the characterization of subsequent pregnancies as high risk, due to the risk of prematurity and low birth weight in subsequent pregnancies [33,34,35,36], which requires appropriate follow-up [37]. Possibly aware of this, women with this history were more likely to seek care.

Premature uterine contractions may be associated with the presence of ZIKV infection, which, upon coming into contact with the maternal immune system and placental/fetal unit, may have triggered the activity of maternal cytotoxic lymphocytes and placental pro-inflammatory activity, which, in turn, may also be responsible for the onset of labor [38,39], leading to unfavorable gestational outcomes, such as prematurity and low birth weight.

The relationship between anthropometric measurements of neonates and maternal ZIKV infection is not statistically significant—which was also observed in a large-scale longitudinal study conducted in Dallas with ZIKV-positive and -negative pregnant women [40]—not excluding the possibility of developmental changes during long-term child growth and development [33]. This reinforces the need for childcare for the children of women with ZIKV infection during the gestational period.

The similarities between the values of blood components and platelets according to the status of maternal infection led us to consider the possibility of false negative results among pregnant women with non-positive results for ZIKV attested by RT-PCR, since they presented symptoms similar to ZIKV-positive pregnant women, but some with a test time of longer than 15 days. The possibility of infection, even with negative results in urine and blood samples tested by RT-PCR, cannot be disregarded, nor can the sensitivity and specificity values of the test, especially in areas endemic for arboviruses [41].

The significant changes in the lymphocyte values of pregnant women with ZIKV infection were the main hematological finding related to infection in pregnant women. The observed cases of hypochromia, leukopenia, neutropenia, and low mean platelet volume (MPV) diagnosed among pregnant women, although not significant, present an overview of possible changes to pregnant women with ZIKV infection. The results contradict the neutrophilia reported in non-human primate pregnant females with ZIKV infection [19].

The maternal hematological profile described in this study may assist health professionals in the characterization of ZIKV infection and the differentiation of other arboviruses in pregnant women, since thrombocytopenia and leukopenia are the main hematological alterations documented for other arboviruses, such as DENV and CHIKV [12,42].

In addition, the findings comprise maternal hematological alterations related to ZIKV infection during pregnancy. The scientific literature reviewed offers an overview related only to the general population, describing leukocytosis [15], leukopenia and monocytosis [18], thrombocytopenia [13,14,16,17,18], and the development of idiopathic thrombocytopenic purpura [43].

The association between the increase in platelet count and the occurrence of an unfavorable gestational outcome indicates that important changes in blood component values during pregnancy can trigger deleterious outcomes. Throughout the gestational period, the amount of platelets circulating in the maternal bloodstream is expected to be low due to placental consumption [44,45]. Miscarriage and fetal losses are described as among the main outcomes of pregnant women with platelet disorders, as in cases of essential thrombocythemia [46].

The increase in circulating platelets in the maternal bloodstream may be related to changes in placental activity in pregnant women with infections. The presence of infections in the maternal organism may contribute to failures in the process of angiogenesis and placental vasculogenesis, which can interfere with the development of an adequate pregnancy, resulting in miscarriage, restricted intrauterine growth, premature births, and low birth weight [47].

The present study allowed us to identify maternal hematological alterations resulting from ZIKV infection and the repercussions of changes in blood components for gestational outcomes. However, the study possesses limitations inherent to an observational clinical study of routine follow-ups. The fact that most pregnant women performed only one blood count during the follow-up in the outpatient clinic did not allow for observation of the temporality and evolution of maternal hematological characteristics throughout the gestational and postpartum period. Despite the planned follow-up, the women joined the study at different stages of pregnancy and were free to return to subsequent consultations or not, which hindered the uniformity in the number of consultations and examinations of the women followed up.

The results presented here help in the expansion of knowledge related to maternal hematological repercussions resulting from ZIKV infection, by carefully evaluating the results of complete blood counts of pregnant women according to their infection status, describing the hematological characteristics found, and taking into account the changes that frequently occur during pregnancy.

## 5. Conclusions

This study described the hematological profile of pregnant women with an exanthematic condition reported during the most intense phase of ZIKV transmission in Amazonas. There were significant changes in the lymphocytic values of ZIKV-positive pregnant women. There were no associations between maternal hematological profile and socio-demographic, obstetric, or infectious characteristics. The study showed that the increase in maternal platelet count is associated with the occurrence of unfavorable gestational outcomes.

In addition to ZIKV infection, maternal hematological alterations may contribute to the occurrence of deleterious outcomes. It is necessary to offer adequate follow-up during prenatal care, in order to identify in a timely manner any changes, the presence of infections, and the prescription of treatment in order to avoid complications to maternal and child health.

## Figures and Tables

**Figure 1 viruses-13-00710-f001:**
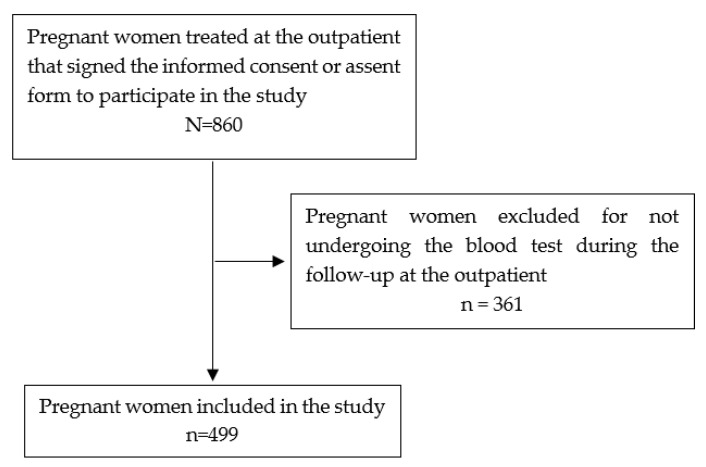
Inclusion flow of pregnant women participating in the study, Manaus, 2020.

**Table 1 viruses-13-00710-t001:** Gestational outcome according to the infection status of pregnant women with exanthematic manifestation who sought care at the Foundation of Tropical Medicine Doctor Heitor Vieira Dourado, Manaus, Brazil.

Variables	Zika Virus	Zika Virus and Other Infection *	Other Infec tions *	Non-Positive
	*n* = 166 (%)	*n* = 46 (%)	*n* = 57 (%)	*n* = 230 (%)
Miscarriage	4 (2.4)	0 (0.0)	1 (1.7)	0 (0.0)
Fetal death	1 (0.6)	1 (2.1)	0 (0.0)	4 (1.7)
Prematurity	17 (10.2)	1 (2.1)	3 (5.2)	19 (8.2)
Low birth weight	14 (8.4)	2 (4.2)	2 (3.5)	11 (4.8)
Microcephaly	6 (3.6)	0 (0.0)	0 (0.0)	1 (0.4)
Apgar score < 7 at the first minute of life	11 (6.6)	1 (2.1)	4 (7.0)	9 (3.9)

Note: * Other infections: Dengue, Chikungunya, Mayaro, HIV, Hepatitis B, Syphilis, Toxoplasmosis, Rubella, Cytomegalovirus, Herpes, Mononucleosis, and Parvovirus.

**Table 2 viruses-13-00710-t002:** Hematological variables according to the infection status of pregnant women with exanthematic manifestations who sought care at the Foundation of Tropical Medicine Doctor Heitor Vieira Dourado in Manaus, Brazil.

Variables	Infection Status				*p*-Value
	Zika Virus	Zika Virus and Other Infection *	Other Infections *	Non-Positive	
	*n* = 166	*n* = 46	*n* = 57	*n* = 230	
	Median (IQR)	Median (IQR)	Median (IQR)	Median (IQR)	
Hemoglobin					0.773
Mean (SD)	12 (1.1)	12.2 (1)	12 (0.9)	12.1 (1.1)	
Hematocrit					0.435
Mean (SD)	35.3 (3.3)	36.1 (3.2)	35.5 (2.9)	35.3 (3.2)	
MCV	88.6 (85.6; 92.2)	89.5 (85.8; 91.7)	87.7 (84.2; 91.5)	88.2 (85.5; 90.8)	0.482
MCH	30.2 (29.2; 31.5)	30.1 (28.8; 31.2)	29.9 (28.4; 31.1)	30.1 (29; 31.1)	0.494
MCHC	34.1 (33.4; 34.7)	33.8 (33.2; 34.6)	33.7 (33.3; 34.6)	34.1 (33.7; 34.6)	0.154
RDW	13.2 (12.3; 14)	13.2 (12.2; 14.2)	12.9 (11.8; 14)	12.9 (12.1; 13.9)	0.261
Leukocytes	**b ****	**ab**	**a**	**a**	0.006
	6350 (5332.5; 7577.5)	6550 (5005; 7925)	7100 (6210; 8200)	7050 (5500.9045)	
Segmented	4516.5 (3598; 5629.8)	4788 (3640; 5715)	4884 (3600; 5751)	5008 (3689.8; 6122.5)	0.128
Eosinophils	**ab**	**b**	**ab**	**a**	0.009
	88 (53.5; 149.2)	73 (54; 125)	98 (66; 158)	114.5 (62; 194.2)	
Basophils	**ab**	**ab**	**a**	**b**	0.007
	15 (0.24)	9.5 (0; 20.8)	16 (0; 33)	5 (0.23)	
Lymphocytes	**b**	**b**	**a**	**a**	**<0.001**
	1299 (966; 1681.5)	1254 (1001; 1689)	1620 (1329; 2047)	1487.5 (1134; 1953.5)	
Monocytes	380.5 (319.8; 480)	396 (314; 516)	465 (352; 567)	425 (319.8; 585.5)	0.069
Platelet count	219,500 (189,500; 259,500)	247,500 (216,950; 275,250)	241,000 (189,000; 287,000)	239,000 (194,250; 284,750)	0.131
MPV	8.5 (7.8; 9.1)	8.6 (7.7; 9.4)	8.6 (7.9; 9.1)	8.6 (7.9; 9.2)	0.624
PCT	0.2 (0.2; 0.2)	0.2 (0.2; 0.2)	0.2 (0.2; 0.2)	0.2 (0.2; 0.2)	0.058
PDW	17.1 (16; 17.9)	17.2 (16.5; 17.5)	16.9 (15.4; 17.8)	16.8 (14.5; 17.4)	0.102

IQR: Interquartile range, SD: Standard deviation, MCV: Mean corpuscular volume, MCH: Mean corpuscular hemoglobin, MCHC: Mean corpuscular hemoglobin concentration, RDW: Red cell distribution width, MPV: Mean platelet volume, PCT: Procalcitonin, PDW: Platelet distribution width. * Other infections: Dengue, Chikungunya, Mayaro, HIV, Hepatitis B, Syphilis, Toxoplasmosis, Rubella, Cytomegalovirus, Herpes, Mononucleosis, and Parvovirus. ** Dunn’s nonparametric comparison test. Different letters differ at the 0.05 significance level.

**Table 3 viruses-13-00710-t003:** Hematological variables according to the gestational outcome of pregnant women with exanthematic manifestations who sought care at the Foundation of Tropical Medicine Doctor Heitor Vieira Dourado, Manaus, Brazil.

Variables	Gestational Outcome		*p*-Value
	Unfavorable *	Favorable	
	*n* = 84	*n* = 388	
	Median (IQR)	Median (IQR)	
Hemoglobin			0.664
Mean (SD)	12 (1.1)	12 (1)	
Hematocrit			0.463
Mean (SD)	35.1 (3.2)	35.4 (3.1)	
MCV	87.8 (84.2; 90.6)	88.5 (85.6; 91.6)	0.077
MCH	30.1 (28.5; 30.8)	30.1 (28.9; 31.3)	0.329
MCHC	34.1 (33.4; 34.7)	34.1 (33.5; 34.6)	0.813
RDW	13.2 (12.1; 13.9)	13.1 (12.1; 14)	0.763
Leukocytes	6600 (5427; 5.8440)	6800 (5575; 8137.5)	0.644
Segmented	4790 (3451.2; 5943)	4769 (3699.8; 5808.2)	0.635
Eosinophils	105 (58.5; 160.8)	96.5 (56; 162)	0.905
Basophils	14 (0; 26.2)	11 (0; 24)	0.61
Lymphocytes	1442.5 (1125; 1866.5)	1424 (1088.2; 1901.2)	0.661
Monocytes	438.5 (328.8; 509.2)	413 (323.2; 551.5)	0.888
Platelet count	244,000 (208,750; 297,250)	227,000 (189,000; 276,250)	0.024
MPV	8.5 (7.7; 9.2)	8.6 (7.8; 9.2)	0.567
PCT	0.2 (0.2; 0.3)	0.2 (0.2; 0.2)	0.05
PDW	17 (14.5; 17.8)	17 (15; 17.6)	0.819

IQR: Interquartile range, SD: Standard deviation, MCV: Mean corpuscular volume, MCH: Mean corpuscular hemoglobin, MCHC: Mean corpuscular hemoglobin concentration, RDW: Red cell distribution width, MPV: Mean platelet volume, PCT: Procalcitonin, PDW: Platelet distribution width. * Unfavorable outcome: miscarriage, fetal and/or neonatal death, prematurity, low birth weight, microcephaly, and immediate neonatal adaptation using the Apgar index ≤ 07 points during the first minute of life.

**Table 4 viruses-13-00710-t004:** Relative risk for unfavorable gestational outcome pregnant women with exanthematic manifestations who sought care at the Foundation of Tropical Medicine Doctor Heitor Vieira Dourado, Manaus, Brazil.

Predictors	Unfavorable Gestational Outcome		
	Relative Risk	CI (95%)	*p*-Value
(Intercept)	0.059953	0.029551–0.121632	**<0.001**
Platelet count *	1.000004	1.000002–1.000007	**0.001**

Note: * Number of observations; 396 results.

## Data Availability

The data presented in this study are available on request from the corresponding author.

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
