# Peer review of "Hematological Profile of Pregnant Women with Suspected Zika Virus Infection Followed Up at a Referral Service in Manaus, Brazil"

_viruses, 2021, doi:10.3390/v13040710_

Round 1

Reviewer 1 Report

Dear authors,

The scientific community and overall society appreciate your contribution to science.

Following my comments:

Line 62 - "FMT-HVD is a reference center for the treatment of acute, infectious, parasitic, and dermatological febrile diseases."

Is FMT-HVD a National reference center? Regional reference center? Please clarify.

Line 72 - Replace "urinary" by "urine"

Line 75 - 

"For blood samples, 10 mL was collected through venipuncture on the forearm of each pregnant woman using a system with a sterile and disposable needle. Each sample was stored in a sterile tube and subsequently analyzed."

What kind of blood tubes? With anticoagulant? What kind?

Line 79 - "Using the blood samples, the presence of infections that make up enlarged TORCH syndrome (toxoplasmosis, rubella, syphilis, cytomegalovirus, infectious herpes mononucleosis, and parvovirus) was investigated, in addition to chronic infections of hepatitis B and C. The presence of HIV infection was investigated using an immuno-enzymatic assay (anti-HIV—ELISA I).

Replace by:

"Blood samples were preliminarily tested for agents involved in the TORCH syndrome which is a cluster of symptoms and clinical signs caused by congenital infection of viruses as rubella virus, cytomegalovirus, herpes simplex parvovirus, and Varicella zoster, as well as other agents as Toxoplasma and syphilis."

How these agents, including hepatitis B and C were tested? ELISA? PCR? Please include that information for each agent to the text.

Line 83 - "For the diagnosis of ZIKV, blood and urine samples were processed through the reverse transcriptase reaction followed by real-time polymerase chain reaction (RT-PCR), a method employed by the Central Laboratory of Public Health (LACEN) of Manaus and Laboratory of Clinical  Analysis (LAC) of FMT-HVD, which were the laboratories responsible for performing the ZIKV test."

Please, add what was or were the protocols of RT-PCR for ZIKV used including their references or information if kits. I understand the authors may have not ran these tests, but this information should be added here.

Line 88 - "Diagnosis of infection by the other arboviruses occurred through detection of viral RNA through RT-PCR, and if the patient had been manifesting the disease for five or more days, the samples were evaluated by immuno-enzymatic assays."

What arboviruses? DENV and CHIKV? Please add this information here. As suggested for ZIKV RT-PCR, please cite the kit, or the reference related protocol used for testing these other arboviruses. 

Line 90 - "Due to the impossibility of performing serologies in pregnant women, the examinations performed during prenatal care were consulted, which had already been performed at LACEN during the prenatal routine."

Why performing serologies in pregnant women was not possible? What examinations the authors are referring to? This sentence is hard to understand. Please, reword it to clarify.

Line 103 - Replace: "by an Apgar Score ≤ 7 points during the first minute of life."

By: "by an Apgar Score ≤ 7 points, which is a test given at 1 minute after birth to check a baby's heart rate, muscle tone, and other signs to see if extra medical care or emergency care is needed."

Line 171 - "There were significant associations between the values of Mean Corpuscular Volume, Platelet Count, Procalcitonin, and unfavorable gestational outcomes"

Please, clarify why these parameters had significant association considering their P-value and median.

Table 3 - Is the P-value of PCT accurate? 

Author Response

Dear reviewer, thank you for the comments. 

Best regards. 

Reviewer 2 Report

This reviewed version has improved significantly. I'm comfortable to recommend its publication now.

Author Response

Dear Reviewer,

Thank you for the comments. 

Best regards.

This manuscript is a resubmission of an earlier submission. The following is a list of the peer review reports and author responses from that submission.

Round 1

Reviewer 1 Report

The authors have investigated whether infection of pregnant women with ZIKV results in specific hematological changes. They report that ZIKV, through unknown mechanisms, may result in a significant low blood lymphocyte count and high platelet count in pregnant women. The authors demonstrate that the high platelet count was associated with unfavourable gestational outcome. The study outcome represents important clinical implications for the establishment of updated medical protocols, and for clinicians to deliver more informed services to pregnant patients with ZIKV infection.

Reviewer 2 Report

This manuscript has the main objective to describe the hematological profile of pregnant women with suspected Zika virus. The authors enrolled 499 pregnant women on a reference center in Manaus, Brazil.

These women were allocated in four different groups. 

As an overall comment, the English language needs extensive revision. Apparently the manuscript was originally written in Portuguese and than translated literally. There is some sentences that are very difficult to understand. As an example, see the long sentence in lines 38-40. 

 I have some questions about the methodology.  Although the authors say that they enrolled pregnant women, in lines 68-69 they write : "Sociodemographic, obstetric and neonatal data were collected during the first call of pregnant  women. When available, information described in the prenatal and child booklet was collected"  This sounds as a contradiction! How could the authors collect NEONATAL data if the women were pregnant? Was a it then a retrospective study?

In lines 85-89  the authors wrote:  "The diagnosis of infection by the other arboviruses occurred through detection of viral RNA  through RT-PCR, and, if the patient had been manifesting the disease for five or more days, the samples would be evaluated by immunoenzymatic assays. In the impossibility of performing serologies by pregnant women, the examinations performed during prenatal care were consulted, already performed at LACEN in prenatal routine".   Besides the problems with English language, it seems that not every women was tested in the reference center and the authors relied on medical records. 

This apparent heterogeneous testing criteria and enrollment reflects in the reliability of the results. 

The main objective of this work was the description of the hematological profile. However in the Methods section there is no mention of when the blood for the hematological analysis was collected, and not if all hematological analyses were made at the same laboratory.  As some women came to the center after 5 days of the beginning of the symptoms, the hematological profile could be heterogeneous according to the timing in the evolution of the infection

Considering this methodological problems it is very difficult to evaluate the results.